# The Kynurenine Pathway in Healthy Subjects and Subjects with Obesity, Depression and Chronic Obstructive Pulmonary Disease

**DOI:** 10.3390/ph16030351

**Published:** 2023-02-25

**Authors:** Per G. Farup, Håvard Hamarsland, Knut Sindre Mølmen, Stian Ellefsen, Knut Hestad

**Affiliations:** 1Department of Research, Innlandet Hospital Trust, N-2381 Brumunddal, Norway; 2Department of Clinical and Molecular Medicine, Faculty of Medicine and Health Sciences, Norwegian University of Science and Technology, N-7491 Trondheim, Norway; 3Section for Health and Exercise Physiology, Faculty of Social and Health Sciences, Inland Norway University of Applied Sciences, N-2624 Lillehammer, Norway; 4Department of Health and Nursing Science, Faculty of Health and Social Sciences, Inland Norway University of Applied Sciences, N-2418 Elverum, Norway

**Keywords:** kynurenine pathway, obesity, depression, chronic obstructive pulmonary disease, inflammation

## Abstract

Background: Changes in tryptophan metabolism through the kynurenine pathway (KP) are observed in several disorders and coupled with pathophysiological deviations. Methods: This study retrospectively compared the KP in serum in healthy subjects (108) with subjects with obesity (141), depression (49), and chronic obstructive pulmonary disease (COPD) (22) participating in four clinical studies and explored predictors of the changes in the KP metabolites. Results: Compared with the healthy group, the KP was upregulated in the disease groups with high kynurenine, quinolinic acid (QA), kynurenine/tryptophan-ratio and QA/xanthurenic acid-ratio and low kynurenic acid/QA-ratio. Tryptophan and xanthurenic acid were upregulated in the depressed group compared with the groups with obesity and COPD. The covariates BMI, smoking, diabetes, and C-reactive protein explained the significant differences between the healthy group and the group with obesity but not between the healthy group and the groups with depression and COPD, indicating that different pathophysiological conditions result in the same changes in the KP. Conclusions: The KP was significantly upregulated in the disease groups compared with the healthy group, and there were significant differences between the disease groups. Different pathophysiological abnormalities seemed to result in the same deviations in the KP.

## 1. Introduction

The kynurenine pathway (KP) is the main route for metabolising free tryptophan in humans. Deviating tryptophan metabolism is observed in various disorders with common pathophysiological characteristics, such as immune system activation [1,2,3]. Obesity, depression, and chronic obstructive pulmonary disease (COPD) are disorders with changes in the KP [4,5,6,7,8,9,10]. These diseases were chosen because they have shown changes in the KP indicating inflammation, and we had data from ongoing studies that allowed comparisons of the KP. Current discrepancies in our understanding of the KP are coupled with its complex regulation by several interacting factors and the opposing effects of its metabolites. Thus, the disease-related abnormalities are not necessarily congruent, and different pathophysiological disturbances might cause comparable deviations in the KP. We are unaware of comparisons of changes in the KP in different diseases and the pathophysiology related to the changes in the KP in the same study. Having data from several studies analysed with identical methods made it possible to explore differences and get new insight into the KP abnormalities and the causes of the abnormalities.

The primary aim of this study was to compare the KP between healthy subjects and subjects with obesity, depression, and COPD. The secondary aims were to explore differences in the KP between the disorders, to study predictors of the changes in the KP metabolites, and to explore the pathophysiology.

## 2. Materials and Methods

### 2.1. Data, Material and Study Design

This study retrospectively combined data from four clinical studies at Innlandet Hospital Trust, Brumunddal, Norway and Inland Norway University of Applied Sciences, Lillehammer, Norway. Four groups were included in the study: Healthy volunteers (the “Healthy” group) and subjects with obesity (the “Obesity” group), depression (the “Depressed” group), and COPD (the “COPD” group).

The Obesity group: Subjects aged 18–65 years old with morbid obesity (defined as BMI > 40 kg/m^2^ or >35 kg/m^2^ with obesity-related complications) without severe not-obesity-related comorbidity referred to the obesity unit at Innlandet Hospital Trust, Gjøvik, Norway for evaluation of bariatric surgery were included in the “Morbid Obesity—Bio-Psycho-Social disorders” study (MO-BiPS study) [11]. Data from the first visit were used.

The Depressed group: Consecutive subjects above 18 years of age with depression according to ICD-10, F 32–34 spectre were included in a cross-sectional study at the Psychiatric division, Innlandet Hospital Trust, Norway and compared with subjects with unexplained neurological symptoms. Subjects with known infections or inflammatory disorders influencing the KP were not included. The background for the study with subject description is given in a previous paper by Hestad et al. [12]. Depression severity measured with the Beck Depression Inventory-II was 30.1 (12.4) (mean (SD)).

The COPD group: In the Granheim COPD Study (ClinicalTrial ID: NCT02598830), subjects with a diagnosis of stable COPD (GOLD grade II-III, predicted forced expiratory volume in the first second (FEV1) between 30–80%, and forced vital capacity (FVC) < 70% after reversibility testing) and healthy volunteers were included. Exclusion criteria were unstable cardiovascular disease, chronic granulomatous disease, known active malignancy in the last five years, serious psychiatric comorbidity, and musculoskeletal disorders preventing the participant from participating in resistance training. The study’s primary aim was to investigate the combined effects of vitamin D_3_ supplementation and resistance training on a range of biological and health- and performance-related outcomes [13]. Only data from the first visit were used (i.e., prior to the onset of the supplementation and exercise training intervention). The subjects in the Granheim COPD Study with a COPD diagnosis constitute the COPD group in the present study.

The Healthy group included subjects from two of the four studies. In addition to the healthy subjects in The Granheim COPD Study, volunteers from another randomised controlled trial conducted at Inland Norway University of Applied Sciences (ClinicalTrial ID: NCT04279951) were included in the Healthy group. This study aimed to study the effects of n-3 polyunsaturated fatty acids supplementation and resistance training on biological, health- and physical performance-related outcomes. Untrained (less than two sessions of resistance exercise per month and less than three hours of endurance exercise per week) was an inclusion criterion. Only data from the first visit in this study were used.

### 2.2. *Variables*

The following variables from the four studies were used in the analyses: Age (years), sex (biological), body mass index (BMI: kg/m^2^), daily smoking (no/yes), diabetes (no/yes) and C-reactive protein (CRP: mg/L).

The Kynurenine pathway: Tryptophan (Trypt), kynurenine (Kyn), kynurenic acid (KA), quinolinic acid (QA), and xanthurenic acid (XA) were quantified in serum. The ratios Kyn/Trypt (K/T ratio × 1000), KA/Kyn × 1000, KA/QA × 1000, KA/XA and QA/XA were calculated. Protein precipitation of 20 µL human plasma after adding 20 µL internal standard solution (one deuterated substance for each of the analytes) was performed using 60 µL 50% (*w*/*v*) trichloroacetic acid (TCA) in water. After thorough mixing (8 min) and centrifugation (15 min, 4000× *g* at 20 °C), an aliquot of 5 µL was injected from the supernatant into the high-performance liquid chromatography (HPLC) system. HPLC was performed with an Agilent 1260 Infinity liquid chromatograph (Agilent Technologies, Palo Alto, CA, USA) with an Agilent 6460 Triple Quad LC/MS detector. Tryptophan and the metabolites were separated on a 3.0 mm × 100 mm, 2.6 µm, EVO C18 reversed-phase column from Phenomenex (Phenomenex Inc., Torrance, CA, USA). The column temperature was 40 °C, and the gradient was 100% 0.1% formic acid in water to 100% 0.1% formic acid in acetonitrile. A seven-point calibration curve was created for each analyte for quantification of Trypt and the metabolites in test samples and control samples. VITAS AS, Oslo, Norway, performed the KP analyses.

### 2.3. Statistics

Descriptive results are given as mean (SD) or number (%). One-Way ANOVA and Chi-square tests were used for unadjusted comparisons between the groups. Differences in the KP metabolites between the groups were analysed with a univariate general linear model with the KP metabolites as dependent variables adjusted for age and sex, and reported as unadjusted regression coefficients (B-values) with 95% confidence intervals.

To study the effect of BMI, smoking, diabetes, and CRP on the KP metabolites, the variables were added one at a time to the first multivariable analyses.

The last analytic step included all variables (age, sex, BMI, smoking, and CRP) at the same time in the multivariable analyses to study fully adjusted differences in the KP metabolites between the groups. Due to missing data, these analyses included 304 subjects. Diabetes as a variable was excluded from this step due to missing information in the COPD group.

IBM SPSS Statistics for Windows, version 27.0 IBM Corp: Armonk, NY, USA, was used for the analyses. Due to the explorative design and multiple testing, *p*-values ≤ 0.01 were judged as statistically significant.

## 3. Results

### 3.1. Participants’ Characteristics

Three hundred and twenty subjects were included in the analyses: 108 in the Healthy group, 141 in the Obesity group, 49 in the Depressed group, and 22 in the COPD group. Table 1 gives the participant characteristics, the number of subjects in each group, and comparisons between the groups. There were statistically significant differences between the groups.

### 3.2. The KP Metabolites and Differences between the Groups

The serum levels of the KP metabolites in the Healthy group and the differences in the KP metabolites between the Healthy group and the groups with diseases are given in Table 2. There were significant positive associations between age and Kyn, KA, and K/T ratio, and between male sex and Trypt and XA. Out of 30 comparisons between the Healthy group and the groups with disease, 11 were statistically significant, showing significant upregulation of the metabolites. The differences between the disease groups showed a significant rise of several metabolites in the Depressed group compared with the two other groups. These results are given in Table 3.

### 3.3. The Associations between the KP and the Predictors BMI, Smoking, Diabetes, and CRP

Table 4 gives the associations between the KP metabolites and one-by-one of BMI, smoking, diabetes and CRP adjusted for age, sex, and disease groups. The KP metabolites not associated with the independent variables are left out. There were significant positive associations between BMI and KA and QA; and between CRP and Kyn, QA, K/T ratio, and QA/XA ratio. There were negative associations between smoking and QA; between diabetes and KA/XA ratio; and between CRP and KA/QA ratio.

### 3.4. Differences in the KP between the Groups of Participants after Adjusting for Age, Sex, BMI, Smoking and CRP, All at the Same Time

Below are given the changes in the differences in the KP between the groups of participants presented in Table 2 and Table 3 after adjusting for age, sex, BMI, smoking and CRP, all at the same time. Only the major and, in principle, different results are given.
Obese compared with Healthy: Kyn: 49 (−13; 112); *p* = 0.120. KA: 1.1 (−1.7; 3.9); *p* = 0.448. QA: 20 (1.6; 39); *p* = 0.034. K/T ratio: 2.9 (−1.8; 7.6); *p* = 0.229. QA/XA ratio: 2.2 (−4.4; 8.7); *p* = 0.510.Depressed compared with Healthy: K/T ratio: 6.4 (2.3; 10.6); ***p* = 0.002**.COPD compared with Healthy: No changes.Depressed compared with Obese: Trypt: 1312 (−95; 2719); *p* = 0.067. XA: 0.55 (−0.08; 1.2); *p* = 0.086.Depressed compared with COPD: QA/XA ratio: −10 (−19; −1.9); *p* = 0.017.Obese compared with COPD: QA/XA ratio: −13 (−23; −4); ***p* = 0.006.**

## 4. Discussion

The most important findings were the differences in the KP between the Healthy group and the groups with a disease, and the influence of the covariates BMI, smoking, diabetes, and CRP on the KP.

### 4.1. Differences in the KP between the Healthy Group and the Groups with Diseases (Section 3.2, Table 2)

There were significant differences between the Healthy group and the groups with diseases. Eleven out of 30 comparisons of the KP metabolites between the Healthy group and the groups with diseases were statistically significant (defined as *p* ≤ 0.01), which is far above the number expected to occur by chance. No KP metabolite showed significant differences between the Healthy group and all disease groups. Nevertheless, there were some overall tendencies: Kyn, QA, K/T-ratio, and QA/XA-ratio were high, and KA/QA ratio was low in the diseased groups compared with the Healthy group. These changes indicate an unfavourable upregulation of the KP in all diseased groups independent of disease. Kyn, K/T-ratio, QA and QA/XA are markers of inflammation, QA is neurotoxic and associated with depression and improves glucose control, KA/XA-ratio has been associated with mood disorders, and KA, QA, and XA regulate the carbohydrate metabolism [1,14,15,16,17,18].

The elevated levels of Kyn, KA, K/T ratio and QA in the Obesity group are in accordance with other studies [6,9,19]. An overexpression of indolamine 2,3-dioxygenase (IDO1) in the adipocytes results in excess Kyn followed by changes in the other metabolites [9]. Reducing the Kyn accumulation protects mice from obesity and could be a target for intervention against obesity [9].

The changes in the Depressed group are in part identical to those in the Obese group with an upregulation of Trypt, Kyn, QA, and XA. The changes differ from the downregulation of Trypt and Kyn in subjects with depression reported in meta-analyses [4,5,20]. The reason for the discrepancy is unknown. However, the relationship between depression and Trypt and Kyn is conflicting. An increase in Kyn and the Kyn to QA pathway is part of a proinflammatory reaction. This increase is seen after interferon treatment [21]. After interferon treatment, antidepression free subjects developing major depression showed a significant increase in Kyn and neopterin, and a more prolonged decrease in Trypt concentrations than those without depression. In our study with severely depressed subjects, Trypt, Kyn, QA, and XA were significantly higher in the Depressed group than in the Healthy group. In subjects with depression, there is also an anti-inflammatory response with a rise in cortisol which might explain the high level of Trypt in the Depressed group [22]. The upregulated QA in this study is in accordance with a meta-analysis [4]. QA, which could be a major contributor to depression, is associated with neuroinflammation, depression, and neurodegeneration. Raised values have been observed together with inflammatory markers in several neurodegenerative diseases [10,15].

The differences between the Healthy and COPD groups seem less impressive than between the Healthy group and the other groups. However, the findings in the COPD group should be interpreted cautiously because of the low number of subjects. Other studies have shown elevated levels of Kyn in subjects with COPD and hypothesised reduced Kyn clearance in the muscles as the pathophysiological mechanism [7,8]. If correct, the pathophysiology explaining elevated Kyn in the COPD group differs from the elevated Kyn in the Obesity group, which is due to the excess Kyn from the adipocytes [9].

The increase in the KP metabolites with age and male sex and the lower levels in smokers are in accordance with other studies [19,23].

### 4.2. Differences in the KP between the Groups with Diseases (Section 3.2, Table 3)

The differences between the groups with diseases were less distinct than between the Healthy group and the groups with diseases. The most unfavourable changes in the KP pathway were in the Depressed group. Trypt and XA were raised in the Depressed group compared with the Obese and COPD groups. The raised Trypt level was not expected [4,5,20]. XA has a crucial role in the control of the dopaminergic activity in the brain, and hence neuropsychological functions [17]. Since QA is a marker of depressive and neurodegenerative disease, the significantly raised QA/XA in the COPD group compared with the Healthy and Depressed groups was unexpected [15].

### 4.3. Associations between the KP Metabolites and One-By-One of the Covariates (Section 3.3, Table 4)

CRP was associated with five KP metabolites, BMI with two, and smoking and diabetes with one of the metabolites.

CRP was associated with high Kyn, QA, K/T-ratio and QA/XA-ratio, and low KA/QA-ratio, confirming findings from other studies that these variables are markers of inflammation [1,3]. Proinflammatory cytokines upregulate IDO expression and shunt the tryptophan metabolism toward the inflammatory markers Kyn and QA [1,3]. Kyn is also metabolised to KA, which has been ascribed neuroprotective, anti-inflammatory and diabetogenic effects [10,14].

The association between Obesity and inflammation, as demonstrated in this study with high CRP values in the Obese group, is well known. CRP was associated with high levels of QA and low KA/QA ratio reflecting the proinflammatory and neurotoxic effects of QA and anti-inflammatory effects of KA [15]. Inflammation could explain some of the changes in the KP in the Obesity group.

The negative association between smoking and most of the KP variables is known from other studies [19,23]. Positive associations were expected since smoking-related oxidative stress is associated with inflammation. Smoking has, however, both pro- and anti-inflammatory effects [24,25,26]. Smoking was negatively associated with QA, which has neurotoxic effects [15]. Long-term smoking-induced changes in locus coeruleus are identical to the changes after long-term use of antidepressant treatment [27].

Most KP metabolites are associated with carbohydrate metabolism [14]. Kyn, KA and XA have diabetogenic effects by inhibiting proinsulin and insulin synthesis and are elevated in subjects with diabetes [14,18]. XA has also, in the same way as QA, been associated with improved glucose control [16]. QA/XA ratio is a marker of insulin sensitivity and glucose control [16]. In this study, the diabetes-related changes in the KP were ambiguous.

### 4.4. Differences in the KP between the Healthy Group and the Groups with Diseases after Adjusting for Age, Sex, BMI, Smoking and CRP, All at the Same Time (Section 3.4)

Adjusting for age, sex BMI, smoking, and CRP at the same time unveiled exciting differences between the groups:

Healthy versus Obese. When adjusting for all the covariates, there were no significant differences in KP metabolites between the Healthy and Obese groups. It is likely that inflammation measured as CRP, and to a lesser extent BMI, explain the differences between Healthy and Obese groups.

Healthy versus Depressed. After adjusting for all the covariates, the significant differences in Trypt, Kyn, QA, and XA persisted. In addition, K/T ratio was statistically significant, indicating that these covariates were not crucial for the KP abnormalities in the Depressed group.

Healthy versus COPD. Adjusting for all covariates did not significantly change the associations between the Healthy and COPD groups. They, therefore, did not explain the differences between the groups. In the COPD group, the effect of smoking is unreliable. The subjects were asked about actual smoking. Most subjects with COPD were previous and often heavy smokers who had recently stopped smoking.

Before adjusting for BMI, smoking, and CRP, the differences in the KP between the Healthy group and the groups with diseases had the same pattern. After adjusting for the covariates, there were significant changes. BMI, smoking, diabetes, and CRP explained the KP abnormalities in the Obese group but not in the Depressed and COPD groups. The results might indicate that the pathophysiology behind similar KP abnormalities differs between the diseased groups. Inflammation measured as CRP seems to drive the difference between Healthy and Obese, but not between Healthy and the other disease groups.

### 4.5. Strengths and Limitations

The participants represented consecutive patients and healthy volunteers treated in a general hospital. Except for the differences in inclusion/exclusion criteria, there was no selection of participants in the four clinical studies. The Healthy group, which was a combination of two groups, might be an exception. One part of the Healthy group was healthy controls in the Granheim COPD study and older than the subjects in the Obese and Depressed groups. The other part of the Healthy group was untrained healthy volunteers in a study of the effects of n-3 polyunsaturated fatty acids supplementation and resistance training on skeletal muscle hypertrophy and therefore motivated for physical activity. All KP-metabolites were analysed in serum, kept in a biobank, and analysed in a specialised laboratory with the same method and at the same time. The comparisons of the KP and the pathophysiology in a single study with identical methods are superior to comparisons between independent studies. However, only a selection of the KP metabolites were analysed. Any effect of the unexpectedly high CRP values in the Healthy group is unknown. Other blood tests were analysed at different local laboratories at inclusion in the studies. There was no uniform way of collecting the clinical information, and differences might have occurred. Data on pharmacological therapy was judged as unreliable and not included in the analyses. To adjust for multiple testing, *p*-values ≤ 0.01 were used as statistically significant.

## 5. Conclusions

The study showed significant differences between the Healthy group and the Obese, Depressed and COPD groups. The KP pathway was upregulated in the disease groups compared with the Healthy group, and the differences between the Healthy group and the disease groups had more or less the same pattern. Adjusting for age, sex, BMI, smoking and CRP unveiled exciting differences. Inflammation, measured as CRP, and in part BMI explained the differences between the Healthy group and the Obese group, but not between the Healthy group and the Depressed and COPD groups. As discussed, the cause of elevated Kyn in subjects with COPD and Obesity might differ. The findings might indicate that pathophysiological differences cause similar changes in the KP.

## Figures and Tables

**Table 1 pharmaceuticals-16-00351-t001:** Participant characteristics are given as mean (SD) and number (proportion as %).

Characteristics (No. If Less than 320)	Healthy n = 108	Obese n = 141	Depressed n = 49	COPD n = 22	Statistics (*p*-Values)
Age (years) †	57.9 (11.6)	43.0 (8.7)	45.3 (14.6)	69.3 (5.1)	<0.001 *
Sex (male)	46 (43%)	31 (22%)	23 (47%)	14 (64%)	<0.001 **
BMI (kg/m^2^) (n = 315)	27.6 (5.4)	42.1 (3.8)	25.8 (4.4)	24.7 (5.0)	<0.001 *
Daily smoker (n = 315)	1 (1%)	25 (18%)	23 (48%)	4 (20%)	<0.001 **
Diabetes (n = 232)	1 (2%)	26 (19%)	2 (4%)	N.A.	0.002 **
CRP (mg/L) (n = 313)	2.6 (5.8)	7.1 (6.3)	2.1 (2.6)	5.0 (7.7)	<0.001 *

* One-Way ANOVA. ** Chi-square. † The mean (SD) age of the “Fatty acid/resistance training” group and the COPD group constituting the Healthy group were 47 (7) and 67 (4) years, respectively.

**Table 2 pharmaceuticals-16-00351-t002:** Serum levels of the KP metabolites in the Healthy group and differences in metabolites between the Healthy group and the Obesity, Depressed and COPD groups adjusted for age and sex. The results are given as mean (SD) and unadjusted regression coefficients (B-values) with 95% CI and *p*-values.

Dependent Variables		Independent Variables				
KP Metabolites	KP Metabolites Healthy Group (ng/mL)	Age (Years)	Sex (Male)	Obese Compared with Healthy	Depressed Compared with Healthy	COPD Compared with Healthy
Tryptophan (Trypt)	12379 (1978)	−6 (−32; 19) *p* = 0.626	909 (317; 1500) ***p* = 0.003**	342 (−400; 1084) *p* = 0.365	1606 (705; 2507) ***p* = 0.001**	125 (−1058; 1308) *p* = 0.836
Kynurenine (Kyn)	440 (106)	2.2 (0.9; 3.6) ***p* = 0.002**	17 (−15; 49) *p* = 0.298	85 (45; 125) ***p* < 0.001**	96 (48; 145) ***p* < 0.001**	37 (−26; 101) *p* = 0.248
Kynurenic acid (KA)	12.8 (4.2)	0.11 (0.04; 0.17) ***p* = 0.001**	0.35 (−1.09; 1.78) *p* = 0.635	3.10 (1.30; 4.90) ***p* = 0.001**	1.11 (−1.07; 3.29) *p* = 0.318	−0.42 (−3.28; 2.45) *p* = 0.774
Quinolinic acid (QA)	83 (30)	0.43 (0.006; 0.86) *p* = 0.047	4.9 (−4.9; 14.6) *p* = 0.325	35 (23; 48) ***p* < 0.001**	23 (8; 38) ***p* = 0.003**	15 (−4; 35) *p* = 0.116
Xanthurenic acid (XA)	5.1 (0.9)	0.00 (−0.01; 0.01) *p* = 0.878	0.37 (0.11; 0.63) ***p* = 0.006**	0.18 (−0.15; 0.50) *p* = 0.293	0.70 (0.30; 1.10) ***p* = 0.001**	−0.23 (−0.75; 0.29) *p* = 0.388
K/T ratio × 1000	36 (9)	0.19 (0.08; 0.30) ***p* = 0.001**	−0.93 (−3.4; 1.6) *p* = 0.462	6.1 (2.9; 9.2) ***p* < 0.001**	3.0 (−0.8; 6.8) *p* = 0.118	3.5 (−1.5; 8.5) *p* = 0.169
KA/Kyn ratio × 1000	30 (8)	0.04 (−0.07; 0.14) *p* = 0.504	−0.79 (−3.3; 1.7) *p* = 0.530	0.58 (−2.52; 3.69) *p* = 0.712	−3.69 (−7.46; 0.09) *p* = 0.056	−2.72 (−7.68; 2.24) *p* = 0.281
KA/QA ratio × 1000	170 (70)	0.29 (−0.42; 1.01) *p* =0.423	0.28 (−16.2; 16.7) *p* = 0.973	−23.2 (−43.8; −2.6) *p* = 0.028	−25.6 (−50.6; −0.5) *p* = 0.045	−15.3 (−48.2; 17.6) *p* = 0.361
KA/XA ratio	2.5 (0.9)	0.02 (−0.003; 0.04) *p* = 0.085	0.09 (−0.43; 0.62) *p* = 0.773	0.55 (−0.11; 1.21) *p* = 0.101	−0.11 (−0.91; 0.69) *p* = 0.788	1.52 (0.46; 2.57) ***p* = 0.005**
QA/XA ratio	16.6 (7.2)	0.07 (−0.08; 0.21) *p* = 0.344	0.36 (−2.96; 3.69) *p* = 0.830	6.4 (2.2; 10.5) ***p* = 0.003**	1.6 (−3.5; 6.7) *p* = 0.537	12.6 (6.0; 19.3) ***p* < 0.001**

**Table 3 pharmaceuticals-16-00351-t003:** Differences in the KP metabolites between the Obesity, Depressed and COPD groups adjusted for age and sex. The results are given as unadjusted regression coefficients (B-values) with 95% CI and *p*-values.

KP Metabolites	Depressed Compared with Obese	Depressed Compared with COPD	Obese Compared with COPD
Tryptophan	1263 (440; 2087) ***p* = 0.003**	1481 (82; 2879) *p* = 0.038	217 (−1111; 1546) *p* = 0.748
Kynurenine	11 (−33; 56) *p* = 0.617	58 (−17; 134) *p* = 0.126	47 (−24; 119) *p* = 0.193
Kynurenic acid	−2.0 (−4.0; 0.01) *p* = 0.051	1.5 (−1.9; 4.9) *p* = 0.376	3.5 (0.3; 6.7) *p* = 0.032
Quinolinic acid	−12.7 (−26.3; 0.9) *p* = 0.067	7.1 (−15.9; 30.2) *p* = 0.544	19.8 (−2.1; 41.7) *p* = 0.076
Xanthurenic acid	0.53 (0.16; 0.89) ***p* = 0.005**	0.93 (0.31; 1.55) ***p* = 0.003**	0.41 (−0.18; 0.99) *p* = 0.176
K/T ratio × 1000	−3.1 (−6.5; 0.4) *p* = 0.086	−0.4 (−6.3; 5.4) *p* = 0.876	2.6 (−3.0; 8.2) *p* = 0.367
KA/Kynurenine ratio × 1000	−4.3 (−7.7; −0.8) *p* = 0.015	−1.0 (−6.8; 4.9) *p* = 0.746	3.3 (−2.2; 8.9) *p* = 0.243
KA/QA ratio × 1000	−2.4 (−25.3; 20.5) *p* = 0.838	−10.3 (−49.1; 28.6) *p* = 0.604	−7.9 (−44.8; 29.0) *p* = 0.675
KA/XA ratio	−0.7 (−1.4; 0.1) *p* = 0.076	−1.6 (−2.9; −0.4) *p* = 0.011	−1.0 (−2.1; 0.2) *p* = 0.110
QA/XA ratio	−4.8 (−9.4; −0.14) *p* = 0.043	−11.0 (−18.9; −3.2) ***p* = 0.006**	−6.3 (−13.7; 1.2) *p* = 0.100

**Table 4 pharmaceuticals-16-00351-t004:** Associations between the KP metabolites and one at a time of the variables BMI, smoking, diabetes, and CRP adjusted for age, sex and groups of participants. Only *p*-values ≤ 0.05 are presented. The statistically significant associations (*p* values ≤ 0.01) are in boldface.

Dependent Variables	Independent Variables			
KP Metabolites	BMI (kg/m^2^)	Smoking	Diabetes	CRP
Tryptophan (Trypt)				−51 (−99; 3) *p* = 0.038
Kynurenine (Kyn)		−46 (−90; −3) *p* = 0.037		3.7 (1.1; 6.3) ***p* = 0.005**
Kynurenic acid (KA)	0.21 (0.05; 0.36) ***p* = 0.008**		−3.1 (−5.7; −0.6) *p* = 0.014	
Quinolinic acid (QA)	1.6 (0.6; 2.6) ***p* = 0.003**	−21 (−35; −9) ***p* = 0.001**		1.3 (0.5; 2.1) ***p* = 0.001**
K/T ratio × 1000	0.3 (0.02; 0.5) *p* = 0.037	−4.1 (−7.5; −0.7) *p* = 0.019		0.55 (0.36; 0.75) ***p* < 0.001**
KA/Kyn × 1000				−0.2 (−0.4; 0.03) *p* = 0.025
KA/QA ratio × 1000				−1.8 (−3.1; −0.5) ***p* = 0.007**
KA/XA ratio			−0.6 (−1.1; −0.2) ***p* = 0.006**	
QA/XA ratio	0.42 (0.06; 0.77) *p* = 0.022	−5.5 (−10.0; −0.9) *p* = 0.018	−3.8 (−7.2; −0.4) *p* =0.031	0.37 (0.10; 0.64) ***p* = 0.007**

## Data Availability

The raw data sets generated and analysed during the current study are not publicly available in order to protect participant confidentiality. Case report forms (CRFs) on paper are safely stored. The data were transferred to SPSS for statistical analyses, and the data files are stored by Innlandet Hospital Trust, Brumunddal, Norway, on a server dedicated to research. The security follows the rules given by The Norwegian Data Protection Authority, P.O. Box 8177 Dep. NO-0034 Oslo, Norway. The data are available on request to the authors.

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
