# Peer review of "The Kynurenine Pathway in Healthy Subjects and Subjects with Obesity, Depression and Chronic Obstructive Pulmonary Disease"

_pharmaceuticals, 2023, doi:10.3390/ph16030351_

Round 1

Reviewer 1 Report

The study investigated the kynurenine pathway (KP) between a healthy population and patients with obesity, depression, and COPD, as well as the potential covariates of the changes in the KP metabolites among these diseases. Results showed that the KP was upregulated in the disease groups. Moreover, significant differences were seen between the depressed group and the groups with obesity and COPD. The covariates BMI, smoking, diabetes, and C-reactive protein accounted for the significant differences between the healthy group and the group with obesity. Authors concluded that different pathophysiological abnormalities may result in the same deviations in the KP. The topic is interesting, but I would suggest addressing the following points before publication.

Major

-Please provide the rationale behind your selection of patients with obesity, depression, and COPD. Did you have specific hypotheses in mind when choosing these groups?

-Authors are suggested to rewrite or restructure the Discussion in a more concise and in-depth way. For example, on page 6, lines 37-39, However, the KP metabolites showed significant differences between the groups with diseases and the Healthy group, with variations between the diseased groups. No KP metabolite showed significant differences between the Healthy group and all disease groups” It might be easier to understand if deleting “the KP metabolites showed significant differences between the groups with diseases and the Healthy group, with variations between the diseased groups”. Additionally, authors mentioned that there were some unexpected results, but did not provide any explanation. Please interpret these results.

-On page 9, lines 131 to 133 and the Conclusion part, authors should be careful to make such a conclusion based on the simple statistical analysis.

Minor

-Please clarify the direction of the alterations (upregulations/downregulations) when appropriate, e.g., in the abstract.

-Study type should be mentioned in the title or abstract, to reflect the retrospective and pooled nature of the study.

-Did authors register the present study? If so, please provide the registration number.

-In Materials and methods, please add details on the inclusion and exclusion criteria.

-Authors mentioned that KP metabolites were quantified in serum, this could be added already in the Abstract.

-For the Obesity, COPD, and healthy controls, data from the first visit were used. What data were used for the Depression group?

-For multivariable analysis, the variables were added one-by-one in section 3.3 and added at the same time in section 3.4. The word “one-by-one” is confusing, it may be better to change it to “one at a time”.

Author Response

Major

Q: Please provide the rationale behind your selection of patients with obesity, depression, and COPD. Did you have specific hypotheses in mind when choosing these groups?

Answer:

It has been added to the Introduction that these diseases were chosen because the disorders have shown changes in the KP metabolism indicating inflammation, and because such comparisons have not been performed in the same study with the same methods, and that we had data from ongoing studies that allowed the comparisons of the KP.

It has been added to the Introduction that this was an explorative study without specific hypotheses.

Q: Authors are suggested to rewrite or restructure the Discussion in a more concise and in-depth way. For example, on page 6, lines 37-39, However, the KP metabolites showed significant differences between the groups with diseases and the Healthy group, with variations between the diseased groups. No KP metabolite showed significant differences between the Healthy group and all disease groups” It might be easier to understand if deleting “the KP metabolites showed significant differences between the groups with diseases and the Healthy group, with variations between the diseased groups”. Additionally, authors mentioned that there were some unexpected results, but did not provide any explanation. Please interpret these results.

Answer

The structure of the Discussion follows the structure of the Results and refers point-by-point to the Results. We do not find it necessary to restructure the Discussion but have changed it according to the reviewer's suggestions.

We have made several minor changes to make the Discussion more concise, as proposed by this reviewer and the other reviewers. We agree that the sentence “However, the KP…” was confusing and have removed it (lines 37-39).

The unexpected results have been explained (lines 85-86)

Q: On page 9, lines 131 to 133 and the Conclusion part, authors should be careful to make such a conclusion based on the simple statistical analysis.

Answer:

To weaken the sentence, the word “might” has been added: “The results might indicate that……” (page 9, line 138 and page 9, line 167)

 Minor

Q: Please clarify the direction of the alterations (upregulations/downregulations) when appropriate, e.g., in the abstract.

Answer:

The directions of the alterations have been clarified in the Abstract.

Q: Study type should be mentioned in the title or Abstract, to reflect the retrospective and pooled nature of the study.

Answer

Study type (data from four clinical studies) and the retrospective, explorative design have been added to the Abstract, Introduction and the “Material and Methods” sections (page 2)

Q: Did authors register the present study? If so, please provide the registration number.

Answer

The The Granheim COPD Study and the study of “n-3 polyunsaturated fatty acids supplementation and resistance training on biological, health- and physical performance-related outcomes” were registered in ClinicalTrials, and the registration numbers are given in the manuscript (page 2). The two other studies were observational studies, not registered in ClinicalTrials and only approved by the Norwegian authorities.

Q: In Materials and methods, please add details on the inclusion and exclusion criteria.

Answer:

Supplementary information about inclusion and exclusion criteria has been added in this paper (Material and Methods, page 2). All details have been reported in other papers that are referred to.

Q: Authors mentioned that KP metabolites were quantified in serum, this could be added already in the Abstract.

Answer:

It has been added to the Abstract that the KP metabolites were quantified in serum.

Q: For the Obesity, COPD, and healthy controls, data from the first visit were used. What data were used for the Depression group?

Answer:

The “Depression study” was a cross-sectional study with only one visit. The word “cross-sectional” has been added to the “Material and Methods” (page 2)

Q: For multivariable analysis, the variables were added one-by-one in section 3.3 and added at the same time in section 3.4. The word “one-by-one” is confusing, it may be better to change it to “one at a time”.

Answer:

One-by-one has been replaced with “one at a time” (page 6, line 11, and page 3 – Statistics)

Reviewer 2 Report

In the present paper, Farup and Co-workers described changes in metabolites belonging to the kynurenine pathway (Trypt, Kyn, KA, QA, and XA) in healthy subjects compared to subjects with obesity, depression, and COPD and between the disease groups.

Minor comments

Introduction: this section is too concise, while a more comprehensive background would be necessary to better understand the study’s importance. In other words, it is well known that in all the pathological conditions explored by the Authors alterations of the KP have been described. So why is it important to compare them in a single study? Note that one part of the Healthy group was healthy controls of the COPD study (no healthy control subjects from Depression and Obesity study), and the other part of the group was healthy volunteers in a different study non included in the paper.

Methods: are clearly described. Is there a reason why only a few metabolites of the KP were measured? For example, 3-HK, ANA, PICA, NICA were not measured. Please explain the rationale of the ratios evaluated (K/T, KA/Kyn, KA/QA, KA/XA, QA/XA). Moreover, in the Methods section, any difference in the evaluated parameters between the Healthy groups from the two different studies should be reported. Finally, data on pharmacological therapies possibly affecting KP (i.e. antidepressants, anti-inflammatory drugs etc) should be reported.

Results: CRP levels in the Healthy group seem high for healthy subjects (mean 2.6 mg/L ± 5.8 (SD)) and this can represent an important bias for the analyses conducted and the data interpretation. How do the authors discuss this data? Also, considering the discussion and differences in KP between the groups after adjusting for age, sex, BMI, smoking, and CRP.

Table legend 1: should be better specified “…the Healthy group were 47 (7) and 67 (4) years, respectively” what is means.

Table 1: include the unit of measurement of CRP.

Discussion: The data are barely discussed and, in some cases, superficially. See below for some examples.

Page 7 from line 74: “… from the elevated Kyn in the Obesity group, which is due to the excess Kyn from the adipocytes”. The sources of Kyn in the latter group can be different.

Page 7 from line 84: “The significantly raised QA/XA in the COPD group compared with the Healthy and Depressed groups was unexpected.” The Authors should attempt to explain the observed data.

Page 7 from line 96: “The association between BMI and high levels of KA and QA might reflect the inflammation associated with obesity. QA has been ascribed pro-inflammatory and neurotoxic effects.” In contrast, KA has been ascribed anti-inflammatory and neuroprotective effects. How do the authors explain this?

Page 7 from line 102: “Smoking was negatively associated with QA, which has neurotoxic effects. Long-term smoking-induced changes in locus coeruleus are identical to the changes after long-term use of antidepressant treatment. It can be questioned if smoking is an antidepressant.” This part of the discussion is very speculative and should be considered whether to remove it. The Authors should be considered the role of alpha-7 regulation by nicotine to explain the effects of smoking in KP for example.

There is no need to include references to the results in the Discussion section.

Author Response

 Introduction: this section is too concise, while a more comprehensive background would be necessary to better understand the study’s importance. In other words, it is well known that in all the pathological conditions explored by the Authors alterations of the KP have been described. So why is it important to compare them in a single study? Note that one part of the Healthy group was healthy controls of the COPD study (no healthy control subjects from Depression and Obesity study), and the other part of the group was healthy volunteers in a different study non included in the paper.

Answer:

  • The Introduction has been enlarged and made clearer. The study compared the KP and the pathophysiology associated with the abnormal KP between healthy subjects and subjects with diseases, and between subjects with different diseases. These differences in the KP and the differences in the pathophysiology made it possible to get new insight into the KP abnormalities and the causes of the abnormalities. We are unaware of such an approach to the study of the KP.
  • The comparisons in a single study with identical methods are superior to comparisons between independent studies. This statement has been added to the “Strengths and limitations” (lines 150-152)
  • The possible limitation related to the control group was discussed in the “Strengths and Limitations” section in the previous version.

 Methods: are clearly described. Is there a reason why only a few metabolites of the KP were measured? For example, 3-HK, ANA, PICA, NICA were not measured. Please explain the rationale of the ratios evaluated (K/T, KA/Kyn, KA/QA, KA/XA, QA/XA). Moreover, in the Methods section, any difference in the evaluated parameters between the Healthy groups from the two different studies should be reported. Finally, data on pharmacological therapies possibly affecting KP (i.e. antidepressants, anti-inflammatory drugs etc) should be reported.

Answer:

  • When planning the study, we presumed that the measured metabolites were the most important ones. It has been added as a limitation that not all KP metabolites were measured (line 153).
  • The ratios were included because at least some of them are commonly used and seem to be better predictors of disease than the individual metabolite. The significance of some of the ratios is mentioned in the manuscript.
  • The main difference between the two parts of the Healthy group was age. This difference in age has been clarified in the legend to Table 1. Any differences between the groups have been adjusted for in the multivariable analyses.
  • Data on pharmacological therapy was unreliable and not included in the paper. This has been added as a limitation (lines 151-152).

 Results: CRP levels in the Healthy group seem high for healthy subjects (mean 2.6 mg/L ± 5.8 (SD)) and this can represent an important bias for the analyses conducted and the data interpretation. How do the authors discuss this data? Also, considering the discussion and differences in KP between the groups after adjusting for age, sex, BMI, smoking, and CRP.

Answer:

  • We agree that CRP in the Healthy group was higher than expected and more like the disease groups. To what extent this is a bias is unknown. The high values might have reduced the differences between the Healthy group and the disease groups, both in the unadjusted and adjusted analyses. A sentence has been added to the “Strengths and limitations” (lines 153-154).

Table legend 1: should be better specified “…the Healthy group were 47 (7) and 67 (4) years, respectively” what is means.

Answer:

  • The sentence has been specified – see comment above (legend to Table 1)

Table 1: include the unit of measurement of CRP.

Answer:

  • The unit (mg/L) has been added (Table 1).

 Discussion: The data are barely discussed and, in some cases, superficially. See below for some examples.

 Page 7 from line 74: “… from the elevated Kyn in the Obesity group, which is due to the excess Kyn from the adipocytes”. The sources of Kyn in the latter group can be different.

Answer:

  • The words “in subjects with COPD” have been added to clarify the meaning (line 73).  

 Page 7 from line 84: “The significantly raised QA/XA in the COPD group compared with the Healthy and Depressed groups was unexpected.” The Authors should attempt to explain the observed data.

Answer:

  • The unexpectedly raised QA/XA ratio in the COPD group has been explained (lines 85-86).

 Page 7 from line 96: “The association between BMI and high levels of KA and QA might reflect the inflammation associated with obesity. QA has been ascribed pro-inflammatory and neurotoxic effects.” In contrast, KA has been ascribed anti-inflammatory and neuroprotective effects. How do the authors explain this?

Answer:

  • The section has been reformulated (lines 98-103).

 Page 7 from line 102: “Smoking was negatively associated with QA, which has neurotoxic effects. Long-term smoking-induced changes in locus coeruleus are identical to the changes after long-term use of antidepressant treatment. It can be questioned if smoking is an antidepressant.” This part of the discussion is very speculative and should be considered whether to remove it. The Authors should be considered the role of alpha-7 regulation by nicotine to explain the effects of smoking in KP for example.

Answer:

  • The sentence about a possible antidepressant effect of smoking has been removed, as proposed by the reviewer. To discuss alpha7 nicotinic receptors and the KP is outside the scope of this paper.

There is no need to include references to the results in the Discussion section.

Answer:

  • The references to the Results in the Discussion were done to increase readability. One of the reviewers commented upon the structure of the Discussion. The references show that the structure of the Discussion follows the structure of the Results and increases the readability.

Reviewer 3 Report

In my opinion, the study should be rejected due to the choice of the control group. A part of the control group is healthy volunteers but following a study on the effect of n-3 polyunsaturated fatty acid supplementation and resistance training on skeletal muscle hypertrophy, and therefore motivated to physical activity. Changing physical activity alters tryptophan metabolism, and this group should not be included in the control group [Saran et al. 2021; DOI: 10.1038/s41598-021-90616-6].

Author Response

In my opinion, the study should be rejected due to the choice of the control group. A part of the control group is healthy volunteers but following a study on the effect of n-3 polyunsaturated fatty acid supplementation and resistance training on skeletal muscle hypertrophy, and therefore motivated to physical activity. Changing physical activity alters tryptophan metabolism, and this group should not be included in the control group [Saran et al. 2021; DOI: 10.1038/s41598-021-90616-6].

Answer:

Reviewer’s comment is probably due to a misunderstanding. All data from the subjects in the study of n-3 polyunsaturated fatty acid supplementation and resistance training on skeletal muscle hypertrophy were collected at inclusion in the study. The participants were healthy subjects with normal daily activity but no regular physical activity/training. Active physical activity/training was an exclusion criterion. A sentence has been added to avoid misunderstanding (page 2).

Reviewer 4 Report

The manuscript is well written,but need some improvement in the introduction by expanding the context of the studies and relevant to the current understanding of the pathway.

It can be accepted. 

Author Response

The manuscript is well written,but need some improvement in the introduction by expanding the context of the studies and relevant to the current understanding of the pathway.

Answer:

Other reviewers also mentioned the weakness of Introduction. The Introduction has therefore been expanded to explain what was explored and why it was of interest.

Round 2

Reviewer 1 Report

Authors revised their manuscript according to the suggestions. I have no further comments

Reviewer 3 Report

Ok after revision,